

# Sequence comparison of the mitochondrial genomes of *Plesionika* species (Caridea: Pandalidae), gene rearrangement and phylogenetic relationships of Caridea

Yuman Sun[1,*], Jian Chen[1,*], Xinjie Liang[1], Jiji Li[1], Yingying Ye[1] and Kaida Xu[2,3]

[1] National Engineering Research Center for Marine Aquaculture, Zhejiang Ocean University, Zhoushan, Zhejiang Province, China

[2] Zhejiang Marine Fishery Research Institute, Zhoushan, Zhejiang Province, China

[3] Key Laboratory of Sustainable Utilization of Technology Research for Fisheries Resources of Zhejiang Province, Scientific Observing and Experimental Station of Fishery Resources for Key Fishing Grounds, Ministry of Agriculture and Rural Affairs, Zhoushan, Zhejiang Province, China

[*] These authors contributed equally to this work.

## ABSTRACT

**Background**. Despite the Caridean shrimps' vast species richness and ecological diversity, controversies persist in their molecular classification. Within Caridea, the Pandalidae family exemplifies significant taxonomic diversity. As of June 25, 2023, GenBank hosts only nine complete mitochondrial genomes (mitogenomes) for this family. The *Plesionika* genus within Pandalidae is recognized as polyphyletic. To improve our understanding of the mitogenome evolution and phylogenetic relationships of Caridea, this study introduces three novel mitogenome sequences from the *Plesionika* genus: *P. ortmanni*, *P. izumiae* and *P. lophotes.*

**Methods**. The complete mitochondrial genomes of three *Plesionika* species were sequenced utilizing Illumina's next-generation sequencing (NGS) technology. After assembling and annotating the mitogenomes, we conducted structural analyses to examine circular maps, sequence structure characteristics, base composition, amino acid content, and synonymous codon usage frequency. Additionally, phylogenetic analysis was performed by integrating existing mitogenome sequences of true shrimp available in GenBank.

**Results**. The complete mitogenomes of the three *Plesionika* species encompass 37 canonical genes, comprising 13 protein-coding genes (PCGs), 22 transfer RNAs (tRNAs), two ribosomal RNAs (rRNAs), and one control region (CR). The lengths of these mitogenomes are as follows: 15,908 bp for *P. ortmanni*, 16,074 bp for *P. izumiae* and 15,933 bp for *P. lophotes*. Our analyses extended to their genomic features and structural functions, detailing base composition, gene arrangement, and codon usage. Additionally, we performed selection pressure analysis on the PCGs of all Pandalidae species available in Genbank, indicating evolutionary purification selection acted on the PCGs across *Pandalidae* species. Compared with the ancestral Caridea, translocation of two tRNA genes, i.e., *trnP* or *trnT*, were found in the two newly sequenced *Plesionika* species—*P. izumiae* and *P. lophotes*. We constructed a phylogenetic tree of Caridea using the sequences of 13 PCGs in mitogenomes. The results revealed that family Pandalidae exhibited robust monophyly, while genus *Plesionika* appeared to be a polyphyletic group.

Corresponding authors
Yingying Ye, yeyy@zjou.edu.cn
Kaida Xu, xkd1981@163.com

**Conclusions**. Gene rearrangements within the Pandalidae family were observed for the first time. Furthermore, a significant correlation was discovered between phylogenetics of the Caridea clade and arrangement of mitochondrial genes. Our findings offer a detailed exploration of *Plesionika* mitogenomes, laying a crucial groundwork for subsequent investigations into genetic diversity, phylogenetic evolution, and selective breeding within this genus.

# INTRODUCTION

Caridea Dana, 1852 is one of the largest infraorders within Decapoda, comprising over 3,400 species distributed among 36 families (*De Grave & Fransen, 2011*; *Liao et al., 2017*). Caridean shrimps, with their wide distribution and varied habitats, provide an excellent model for studying the origin and adaptive evolution of aquatic organisms in different aquatic habitats (*Sun et al., 2020*). Morphological characteristics, such as pereopods and mouthparts, have traditionally been used in taxonomic studies to classify Caridean shrimps, owing to their extensive morphological variation and diverse lifestyles, including free-living and symbiotic relationships (*Felgenhauer & Abele, 1983*; *Xu, Song & Li, 2005*). However, it remains uncertain whether this classification based on morphology reflects the phylogenetic relationships between families and superfamilies (*Ye et al., 2021*). The employment of molecular sequences is aided in reconstructing the phylogenetic relationships among species, effectively addressing the limitations inherent in traditional taxonomy and resolving numerous controversial issues within the fields of classification and systematic evolution (*Wang et al., 2021*; *Wang et al., 2018*; *Miller et al., 2005*; *Bai et al., 2018*). Mitochondrial genomes (mitogenomes), with their simple structure, rich gene content, straightforward extraction, maternal inheritance, high conservation, low mutation rate, and fast gene evolution (*Boore, 1999*), are extensively employed in phylogenetic and phylogeographic analyses across animal taxa (*Gong et al., 2019*; *Elmerot et al., 2002*; *Chak, Barden & Baeza, 2020*). Previous studies have explored the phylogenetic relationships within Caridea using molecular markers, albeit with a limited number of species. Nonetheless, debates regarding its molecular phylogeny persist. Some scholars propose that Atyidae (De Haan, 1849) represents the basal clade of Caridea (*Li et al., 2011*; *Bracken, De Grave & Felder, 2009*), a finding not supported by other studies (*Ye et al., 2021*). Additionally, the monophyly of certain families within Caridea is contentious. *Bracken, De Grave & Felder (2009)* support the monophyly of five families: Alvinocarididae (Christoffersen, 1986), Alpheidae (Rafinesque, 1815), Crangonidae (Haworth, 1825), Pandalidae (Haworth, 1825), and Processidae (Ortmann, 1896) (*Bracken, De Grave & Felder, 2009*). *Li et al. (2011)* report that the majority of Caridea, excluding Hippolytidae (Spence Bate, 1888) and Palaemonidae (Rafinesque, 1815), exhibits monophyly. However, *Ye et al. (2021)*

describe the monophyly of Hippolytidae and Palaemonidae, while *Sun et al. (2020)* specifically describe the monophyly of Palaemonidae. The phylogenetic interrelations between Alvinocarididae and Atyidae further contribute to the systematic complexity within Caridea (*Ye et al., 2021*; *Boore, 1999*; *Li et al., 2011*; *Wang et al., 2019*; *Sun, Sha & Wang, 2021*).

The family Pandalidae Haworth, 1825, is recognized as one of the largest family-level units within the infraorder Caridea, characterized by its species' diverse biological characteristics and lifestyles. This includes the occurrence of protandrous hermaphroditism in *Pandalus* (Leach, 1814) and *Pandalopsis* (Spence Bate, 1888) (*Liao et al., 2019*; *Butler, 1980*; *Komai, 1999*; *Bergström, 2000*), bioluminescence in *Stylopandalus* (Coutière, 1905) and *Heterocarpus* (A. Milne-Edwards, 1881) (*Herring, 1985*), and the ability to form symbiotic relationships with other invertebrates (*Komai, 1999*; *Bruce, 1983*; *Chan, 1991*; *Crosnier, 1997*; *Horká, De Grave & Ďuriš, 2014*). Pandalidae contains 189 species across 23 genera (*De Grave et al., 2009*), which is widely distributed in both shallow and deep waters (*Sun et al., 2020*). Despite the ecological and economic importance of these species, the current mitogenome data available for Pandalidae are rather limited, with only nine complete mitogenomes available in GenBank (until June 25 2023, excluding unverified data) (https://www.ncbi.nlm.nih.gov/nuccore). The genus *Plesionika* Bate, 1888, the most diverse genus in Pandalidae, comprises 93 species (*De Grave & Fransen, 2011*; *Cardoso, 2011*; *Jiang et al., 2018*)) and is widely distributed throughout subtropical and tropical waters worldwide (*Chace & Bruce, 1985*). Up to the present, only two species of the genus have complete mitogenomes available in the GenBank database, namely *Plesionika edwardsii* (OP087601.1) and *Plesionika sindoi* (MH714453.1). Research utilizing the partial mitochondrial sequences (*COI* and *16S rRNA*) indicates *Plesionika* might not form a monophyletic group (*Silva et al., 2013*; *Chakraborty et al., 2015*; *Chakraborty & Kuberan, 2021*). The phylogenetic relationships obtained by *Liao et al. (2019)*, utilizing two partial fragments of mitochondrial (*12S rRNA* and *16S rRNA*) and six nuclear genes (*atp β*, *Enolase*, *H3*, *NaK*, *PEPCK* and *GAPDH*), also indicated that *Plesionika* did not form a monophyletic group. *Silva et al. (2013)* suggested that the deep-water species are paraphyletic with shallow-water species. Their study encompassed a total of seven *Plesionika* species, and phylogenetic analyses were conducted separately using the *16S rRNA* and *COI* genes. Based on the analysis results, these species were classified into two main clades. Clade I mainly consists of species distributed in shallower marine waters (<400 m), including *Plesionika heterocarpus*, *Plesionika scopifera*, and *Plesionika antigai*. In contrast, clade II includes species found in deeper marine waters (>400 m), such as *Plesionika acanthonotus*, *Plesionika narval*, *Plesionika edwardsii*, and *Plesionika martia*. The taxonomic status of various species within the genus *Plesionika* is debatable (*Crosnier, 1997*; *Komai & Chan, 2003*). The expanded availability of complete mitogenomes has the potential to aid in unraveling the phylogeny of *Plesionika*. This can be accomplished by offering multiple loci with varying rates of evolution, thus enhancing our understanding of their evolutionary relationships.

In the present study, we sequenced and analyzed three complete mitogenomes of *Plesionika* species (*i.e.*, *P. ortmanni*, *P. izumiae*, and *P. lophotes*). Our objectives were

**Table 1  Sampling locations and dates for the three samples.**

| Species name | Sampling date | Species location | GenBank |
|---|---|---|---|
| *Plesionika ortmanni* | April 2022 | Zhoushan, Zhejiang Province 122°14′N, 29°97′E | OP650932 |
| *Plesionika izumiae* | April 2022 | Zhoushan, Zhejiang Province 122°14′N, 29°97′E | OP650933 |
| *Plesionika lophotes* | April 2021 | Taizhou, Zhejiang Province 121°43′N, 28°68′E | OP650934 |

to (1) test the hypothesis of non-monophyly of *Plesionika* species; (2) elucidate the taxonomic status of the Pandalidae family within Caridea; (3) investigate mitochondrial gene rearrangement patterns within Caridea; (4) examine the phylogenetic relationships within Caridean shrimps.

# MATERIALS & METHODS

## Sampling, identifcation and DNA extraction

Three wild species of *P. ortmanni*, *P. izumiae* and *P. lophotes* were collected from two different sea areas in Zhejiang Province, China (Table 1). Experts from the Marine Biology Museum of Zhejiang Ocean University morphologically identified the specimens, referencing the literature (*Kim et al., 2012*; *Li, 2006*). The three *Plesionika* species share characteristics such as having no dorsal ridges or protrusions on their abdomens. Additionally, the bristles on the antennal stalk are sharp and pointed, extending to the distal edge of the first antennal segment. The differences include: in *P. ortmanni*, the sixth abdominal segment's length is 1.5 times its maximum height; the tail fan's length is also 1.5 times that of the sixth abdominal segment; three pairs of spines are present on both the dorsal and posterior margins. The length of the antennal scale is 4.3 to 4.4 times its width. The second pair of walking legs is approximately equal in size. In *P. izumiae*, the sixth abdominal segment is 1.7 times its maximum height; the tail fan is 1.4 times the length of the sixth abdominal segment; there are three pairs of small spines on the dorsal margin; and there are three pairs of spines on the posterior margin. The length of the antennal scale is 4.2 times its width. The second pair of walking legs is unequal in size. In *P. lophotes*, the sixth abdominal segment is 1.5 times its maximum height; the tail fan is 1.6 times the length of the sixth abdominal segment; there are four pairs of small spines on the dorsal margin; and there are three pairs of spines on the posterior margin. The length of the antennal scale is approximately 3.4 times its width. The second pair of walking legs is unequal in size. Samples were preserved in absolute ethanol before DNA extraction. The total DNA was extracted using the salt-extraction procedure and stored at −20 °C for sequencing (*Aljanabi & Martinez, 1997*).

## Mitogenomes sequencing, assembly, and annotation

The complete mitogenomes of three *Plesionika* species were sequenced by using next-generation sequencing (NGS) on the Illumina Hiseq X Ten platform by Origin gene Bio-pharm Technology Co. Ltd. (Shanghai, China). The mitochondrial genomic DNA of

the samples underwent initial quality control, where 1.0% agarose gel electrophoresis was utilized to assess the quality of the DNA. Additionally, a nucleic acid quantifier (NanoDrop, Wilmington, DE, USA) was employed to detect the purity and concentration of the DNA. The quality-controlled mitochondrial genomic DNA of the samples was randomly fragmented into 300–500 bp segments using a Covaris M220 ultrasonic disruptor. The fragmented DNA was subsequently purified to construct sequencing libraries. The steps involved are as follows: DNA end repair, 3′ adenylation, sequencing adapter ligation, and recovery of target fragments through agarose gel electrophoresis. The final sequencing libraries were generated through PCR amplification and were subsequently subjected to sequencing on the Illumina HiSeq TM platform. Trimmomatic v0.39 software was used to filter out low-quality reads, duplicate reads, sequences with a high ''N'' ratio, and sequencing adapter sequences (*Bolger, Marc & Bjoern, 2014*). The reads from the three species were reassembled using the NOVOPlasty assembly software (*Dierckxsens, Mardulyn & Smits, 2016*). The assembled sequences were compared with other *Plesionika* species genomes in GenBank, and the *COI* and *16S rRNA* sequences were verified using NCBI BLAST (*Altschul et al., 1997*). Aberrant start and stop codons were identified by comparing with similar codons in other invertebrate species. The online software MITOS (*Bernt et al., 2013*) was utilized for structural and functional annotation, with manual corrections performed to obtain the final complete mitogenome. The sequenced mitogenomes were uploaded to the GenBank database (Table 1).

## Sequence analysis

The circular visualization of the mitogenomes of the three *Plesionika* species was completed using the CGView server (*Grant & Stothard, 2008*). The analysis of the nucleotide composition of the whole mitogenome, protein-coding genes (PCGs), rRNA, tRNA genes, and AT content was conducted using MEGA-X (*Kumar et al., 2018*). The base skew values were calculated using the formulas AT-skew $= (A - T)/(A + T)$ and GC-skew $= (G - C)/(G + C)$ (*Alexandre, Nelly & Jean, 2005*). The accuracy of transfer RNA genes and their secondary structures was confirmed using MITOS (*Bernt et al., 2013*). Base composition, nucleotide composition and relative synonymous codon usage (RSCU) of each protein-coding gene were calculated using MEGA-X (*Kumar et al., 2018*). The Ka/Ks ratios of the three mitogenomes were estimated using DnaSP 6.0 (*Rozas et al., 2017*).

## Gene order analysis

For comparative analyses, an additional 79 complete mitogenomes of Caridea were obtained from GenBank (Table S1), in addition to the three mitogenomes sequenced in this study. The gene arrangements of all 82 mitogenomes were compared with the ancestral Decapoda, with the aim of identifying potential novel gene orders that have not been reported in previous studies. To ensure that observed gene order differences were not caused by mis-annotations, any mitogenome in Caridea that deviated from the ancestral pattern underwent re-annotation using MITOS (*Bernt et al., 2013*).

## Phylogenetic analysis

To explore the phylogenetic relationships of Pandalidae, sequences of 79 species from 11 families within Caridea were downloaded from GenBank (Table S1). The mitogenomes of *Hemigrapsus sinensis* (NC_065995) and *Helicana japonica* (NC_065158) from Brachyura served as outgroup, and phylogenetic analyses were performed based on the 13 PCGs of these 84 species. The sequences of 13 PCGs from each sample were identified using DAMBE 7 software (*Xia, 2018*). The PCG sequences of these 81 species were aligned with MEGA-X's ClustalW (*Kumar et al., 2018*). Subsequently, Gblocks v.0.91b REF was applied to determine and select conservative regions, removing divergent and ambiguously aligned blocks (*Castresana, 2000*). DAMBE 7 was used to assess the suitability of these sequences for phylogenetic tree construction (*Xia, 2018*).

The analysis of phylogenetic relationships was conducted employing maximum likelihood (ML) method through the program IQ-tree 2.1.3 (*Minh et al., 2020*) and Bayesian inference (BI) method through the program MrBayes 3.2.7a (*Ronquist et al., 2012*) to analyze the phylogenetic relationships. The ML tree was constructed using the program IQ-TREE (*Minh et al., 2020*), with the best substitution model (TIM2 + F + R7) being filtered based on Bayesian information criterion (BIC) *via* ModelFinder (*Kalyaanamoorthy et al., 2017*) calculations, and 1,000 ultra-fast bootstraps were set for the rebuilding of the consensus tree. For BI tree construction using MrBayes v3.2, the initial step involved using PAUP 4 (*Swofford, 1993*) for format conversion. This was followed by the integration of PAUP 4, ModelTest 3.7 (*Darriba et al., 2020*) and MRModelTest 2.3 (*Nylander, 2004*) within MrMTgui to identify the optimal alternative model (GTR + I + G) as dictated by the Akaike information criterion (AIC). Four Markov Chain Monte Carlo (MCMC) chains were simultaneously run for 2 million generations, with a sampling frequency of every 1,000 generations. During the initial burn-in phase, 25% of trees were discarded, and convergence of independent runs was evaluated by the mean standard deviation of the splitting frequency (<0.01). Finally, the phylogenetic tree was edited using the software FigTree v1.4.3 (*Rambaut, 2018*).

Both ML and BI methods were used to construct phylogenetic trees. The non-parametric bootstrap support values and Bayesian posterior probabilities generated by these two methods represent the support rates of nodes, respectively. The non-parametric bootstrap method tends to underestimate support rates of nodes, whereas Bayesian method tends to overestimate them (*Suzuki, Glazko & Nei, 2002*). A maximum likelihood value greater than 70% indicates that the clade relationship is well resolved. Conversely, a value between 50% and 70% is considered weak support, and anything below is regarded as unresolved (*Huelsenbeck & Hillis, 1993*). Similarly, a Bayesian posterior probability of 95% or higher indicates that the clade support rate is well established (*Leaché & Reeder, 2002*).

## RESULTS

### Genome structure, composition, and skewness

The three mitogenomes of *Plesionika* species consist of 15,908 bp (*P. ortmanni*), 16,074 bp (*P. izumiae*) and 15,933 bp (*P. lophotes*). The GenBank accession numbers are OP650932,

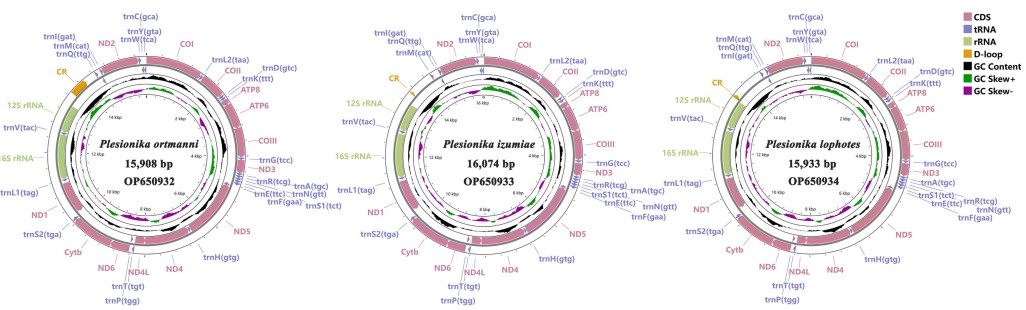

**Figure 1** Complete mitogenome map of three *Plesionika* species.

OP650933 and OP650934, respectively (Fig. 1). These mitogenomes exhibit a closed, circular, double-stranded DNA structure and encompass a total of 37 genes, including 13 PCGs, 22 transfer RNA (tRNA) genes, two ribosomal RNA (rRNA) genes, and a control region (CR). Notably, 14 genes were localized on the light chain, comprising four PCGs (*ND5*, *ND4*, *ND4L*, and *ND1*), eight tRNA genes (*trnF*, *trnH*, *trnP*, *trnL1*, *trnV*, *trnQ*, *trnC* and *trnY*), and two rRNA genes (*16S rRNA* and *12S rRNA*). Conversely, remaining 23 genes were situated on the heavy chain (Fig. 1, Table S2). The CR was located between *12S rRNA* and *trnI* in all three species, with *P. ortmanni* being the longest (472 bp) and *P. izumiae* being the shortest (68 bp) (Table S2).

The nucleotide compositions of these three newly sequenced mitogenomes were: A: 32.70% to 35.91%, T: 31.46% to 31.94%, G: 11.60% to 13.85%, C: 20.55% to 21.98% (Fig. 2A). The contents of A and T exhibited high values, indicating that codon usage was biased towards A and T, which is consistent with the reported complete Pandalidae mitogenomes (*Sun et al., 2020*). The three species had low G and C contents, indicating obvious bias against G and C. Among them, the AT contents ranged from 64.16% to 67.85%, while the AT-skew were positive in the range of 0.019 ~0.058 and GC-skew were negative in the range of −0.227 ~−0.285 (Fig. 2B).

## Protein-coding genes and codon usage

The PCGs in these three *Plesionika* species mitogenomes had total length of 11,192 bp (*P. ortmanni*), 11,134 bp (*P. izumiae*) and 11,041 bp (*P. lophotes*), respectively, including seven NADH dehydrogenases (*ND1-6* and *ND4L*), three cytochrome oxidases (*COI-III*), two ATPases (*ATP6* and *ATP8*) and one cytochrome b (*Cytb*) (Fig. 1, Table S2). Among these species, the *ND5* was identified as the longest PCG, ranging from 1,644 to 1,719 bp, while the *ATP8* gene consistently presented as the shortest, with a uniform length of 159 bp.High AT contents were also observed in the base composition of these species, with *P. lophotes* exhibiting the highest AT content at 66.05%. Additionally, the AT-skew values were found to be negative, ranging from −0.170 to −0.178 (Fig. 2). Upon comparison of initiation and termination codons of all PCGs of the three *Plesionika* species, we found five initiation codons and two termination codons. The PCGs of these three mitogenomes were predominantly initiated with ATG, ATT, and ATA. Exceptions included the *ATP8* of *P. ortmanni* and *ND3* of *P. lophotes*, which started with ATC, and the *COI* of *P. ortmanni*,

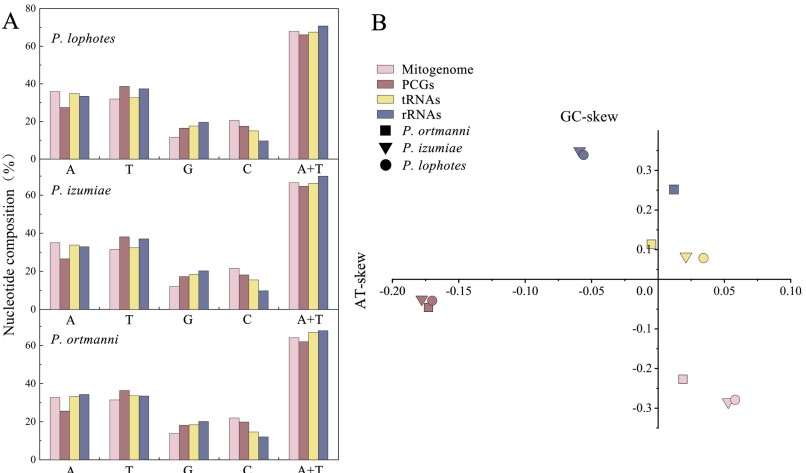

**Figure 2**  Nucleotide composition (A) and nucleotide skews (B) of the three newly sequenced *Plesionika* species mitogenomes.

which commenced with ACG (Table S3). The majority of PCGs in these three mitogenomes terminated with TAA and TAG, while *ND4* in all three mitogenomes, along with *COI* in *P. ortmanni* and *ND5* in *P. lophotes*, terminated with a single T. The occurrence of incomplete termination codons is a notably common phenomenon within the mitochondrial genes of both vertebrates and invertebrates (*Hamasaki et al., 2017*).

The analysis of amino acid compositions in PCGs in the three newly sequenced *Plesionika* species were relatively similar (Fig. 3, Table S4). The most frequently used amino acid is Asn, followed by Leu1, Lys, Phe, Pro, and Thr, while Arg and Cys are less commonly used amino acids. Comparing the relative synonymous codon usage (RSCU) of 13 PCGs in the three species, the result showed that the usage frequency of UUA (Leu), UCU (Ser) and AUA (Met) codons in their sequenced mitogenomes was higher. In *P. ortmanni*, the highest RSCU was found for CGA (Arg), followed by UUA (Leu), ACU (Thr), UCU (Ser) and AUA (Met). In *P. izumiae*, the highest RSCU was found for UUA (Leu), followed by AUA (Met), CAA (Gln), UCU (Ser) and CCU (Pro). In *P. lophotes*, the highest RSCU was found for CGA (Arg), followed by UUA (Leu), UCU (Ser), CCU (Pro) and AUA (Met). The lowest RSCU in all three species was observed for GAG (Glu).

## Transfer and ribosomal RNAs

In common with other Caridea mitogenomes, the mitogenome of three *Plesionika* species contains 22 tRNA genes (Fig. 1, Table S2). The total length oftRNAs in three *Plesionika* species mitogenomes were 1,471 bp (*P. ortmanni*), 1,465 bp (*P. izumiae*) and 1,476 bp (*P. lophotes*), and the length of tRNAs in these species ranging from 59 to 72 bp (Table S2). All tRNAs exhibited high AT contents, with the AT content for the three species being 66.96% (*P. ortmanni*), 66.21% (*P. izumiae*), and 67.48% (*P. lophotes*) (Fig. 2A). The tRNA genes of *P. ortmanni* had a weakly negative AT skew (−0.005) and positive GC skew (0.115), while the tRNA genes of *P. izumiae* and *P. lophotes* had positive AT skew (0.021 and 0.034,

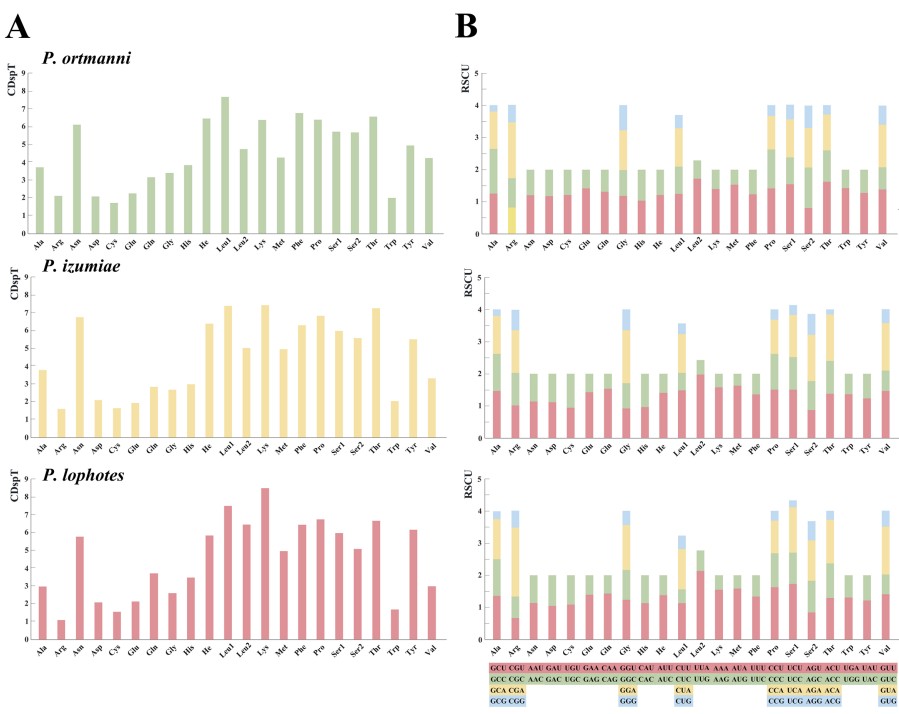

**Figure 3** The frequency of mitochondrial PCG amino acids (A) and relative synonymous codon usage (RSCU) (B) of three newly sequenced *Plesionika* mitogenomes.

respectively) and GC skew (0.083 and 0.079, respectively) (Fig. 2B). The examination of the secondary cloverleaf structure of the 22 tRNAs from these species was conducted. In *P. ortmanni*, it was found that *trnS1* could not form a secondary structure due to the absence of dihydrouracil (DHU) arms, a phenomenon commonly observed in metazoans (*Yamauchi, Miya & Nishida, 2003*) (Fig. 4). Additionally, it was noted that the *trnP* gene in *P. ortmanni*, the *trnA* gene in *P. izumiae*, and the *trnD*, *trnF*, and *trnH* genes in *P. lophotes* were lacking the TΨC loop. In contrast, the remaining genes displayed a typical cloverleaf structure (*Rich & Rajbhandary, 1976*). Comparing the tRNA genes of the three species, it was found that each corresponding amino acid was encoded by the same anticodon.

The total lengths of the *16S rRNA* and *12S rRNA* genes were similar in three species, with *P. ortmanni*, *P. izumiae*, and *P. lophotes* having total lengths of 1,361 bp, 1,328 bp, and 1,322 bp for *16S rRNA*, and 803 bp, 811 bp, and 812 bp for *12S rRNA*, respectively (Table S2). The *16S rRNA* and *12S rRNA* genes of three species were located between *trnL1* and *trnI*, and were separated by *trnV*. High AT contents were also demonstrated, with the AT content for the three species being 67.83% (*P. ortmanni*), 69.98% (*P. izumiae*), and 70.76% (*P. lophotes*) (Fig. 2A). The rRNA genes of *P. ortmanni* had a weakly positive AT skew (0.012) and positive GC skew (0.253), while both the rRNA genes of *P. izumiae* and *P. lophotes* had negative AT skew (−0.059 and −0.056, respectively) and positive GC skew (0.349 and 0.339, respectively) (Fig. 2B).

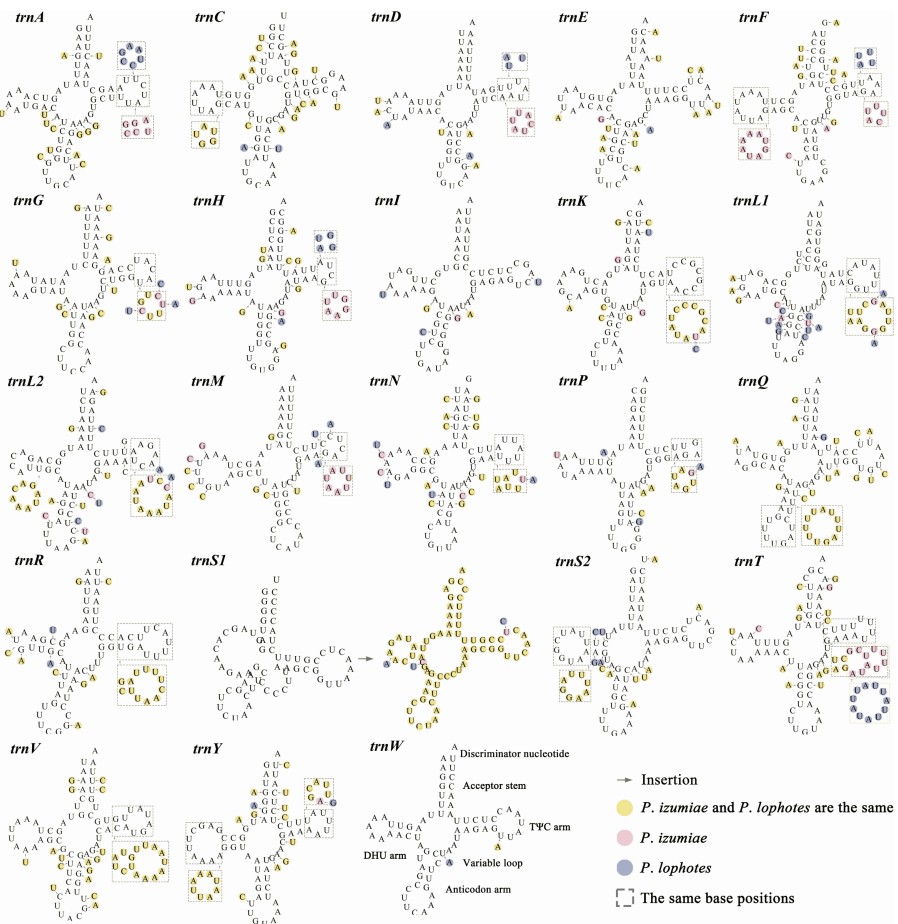

**Figure 4** **The predicted secondary structure of tRNA genes, from *trnA* to *trnW*.** The nucleotide substitution pattern of tRNA genes in three newly sequenced *Plesionika* mitogenomes has been exhibited with the reference species *P. ortmanni*.

## Selective pressure analysis

In genetics, the Ka/Ks ratio, which represents the ratio between the nonsynonymous substitution sites (Ka) and the synonymous substitution sites (Ks), is commonly used to understand the dynamic evolution of PCGs. In this study, the Ka/Ks ratios of the 13 PCGs were calculated using the nine sequenced Pandalidae species (Table S1) to investigate the relationship between evolution and selection pressure (Fig. 5). Results showed that the Ka/Ks ratios of PCGs range from 0.088 (*COI*) to 0.341 (*ATP8*). The Ka/Ks ratio of the *COI* gene was the lowest, indicating that the *COI* gene was under the greatest selection pressure and the gene sequence was relatively conservative. As a result, it is widely used as a potential molecular marker in species identification and phylogenetic studies (*Astrin et al., 2016*).

In general, a gene is considered to be positively selected when the Ka/Ks is greater than 1, neutral evolutionary when the Ka/Ks is equal to 1, and purified selected when the Ka/Ks is less than 1 (*Nei & Kumar, 2000*; *Yang, 2006*). In this study, the Ka/Ks ratios of the 13

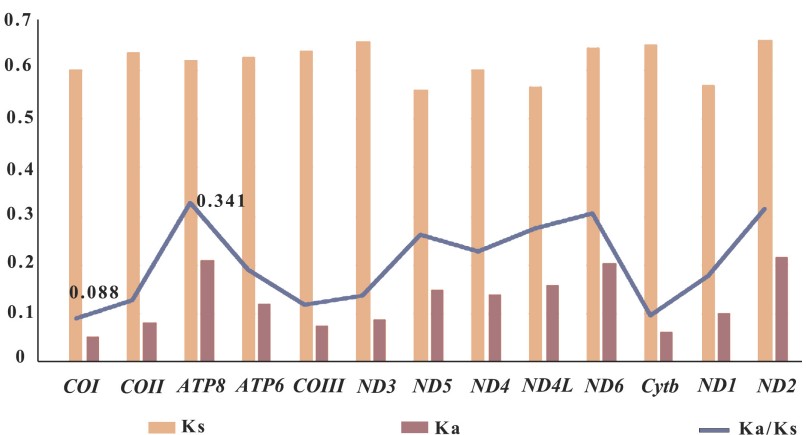

**Figure 5** **Selective pressure analysis for 13 PCGs among 12 Pandalidae mitochondrial genomes.** Species of Pandalidae are shown in Table S1.

PCGs genes were less than 1, indicating that the genes of Pandalidae species were subjected to purification selection during evolution.

## Gene rearrangement

Mitochondrial gene arrangement is an important tool for the study of systematic geography and phylogeny, which provides informative insights into the evolution among metazoans (*Beagley, Okimoto & Wolstenholme, 1998*; *Searle, 2000*). In general, mitochondrial gene arrangement is relatively stable in vertebrates, such as fish, amphibians, and most mammals (*Fu, Chen & Zhao, 2009*). However, in invertebrates, varying degrees of gene rearrangement are commonly observed in mitogenomes (*Ye et al., 2021*; *Boore, 1999*). The gene orders within the infraorder Caridea mitogenomes were compared with those of ancestral Decapoda in this study. It was found that the mitochondrial gene orders (MGOs) of the families Atyidae, Alvinocarididae, Acanthephyridae (Spence Bate, 1888), Oplophoridae (Dana, 1852), and Nematocarcinidae (Smith, 1884) matched those of the ancestral Decapoda. Meanwhile, gene rearrangement was identified in 24 species across 6 families of Caridea (Fig. 6). This contradicts the previous view that the gene order in the Caridea is conservative (*Wang et al., 2018*; *Miller et al., 2005*; *Ivey & Santos, 2007*; *Lü et al., 2019*).

The MGOs of the newly sequenced *P. ortmanni* and the other nine Pandalidae species from GenBank (*i.e., Bitias brevis, Chlorotocus crassicornis, Heterocarpus ensifer, Heterocarpus sibogae, Pandalus borealis, Pandalus prensor, Parapandalus sp., P. edwardsii* and *P. sindoi*) remained consistent with the gene order of ancestral Caridea (*Sun et al., 2020*). Conversely, in the two newly sequenced *Plesionika* species (*P. izumiae* and *P. lophotes*), a translocation has occurred, resulting in a gene order of *trnK-trnD*, as opposed to the ancestral *trnD-trnK* sequence. In Alpheidae, gene rearrangement was observed in *Leptalpheus forceps* and seven *Alpheus* (Fabricius, 1798) species, in which *trnE* translocated and inverted with *trnP* (*Ye et al., 2021*; *Shen et al., 2012*). In addition, *Alpheus lobidens* has an additional *trnQ* repeat downstream of *ND4L* (*Wang et al., 2019*). In Palaemonidae, nine *Palaemon*

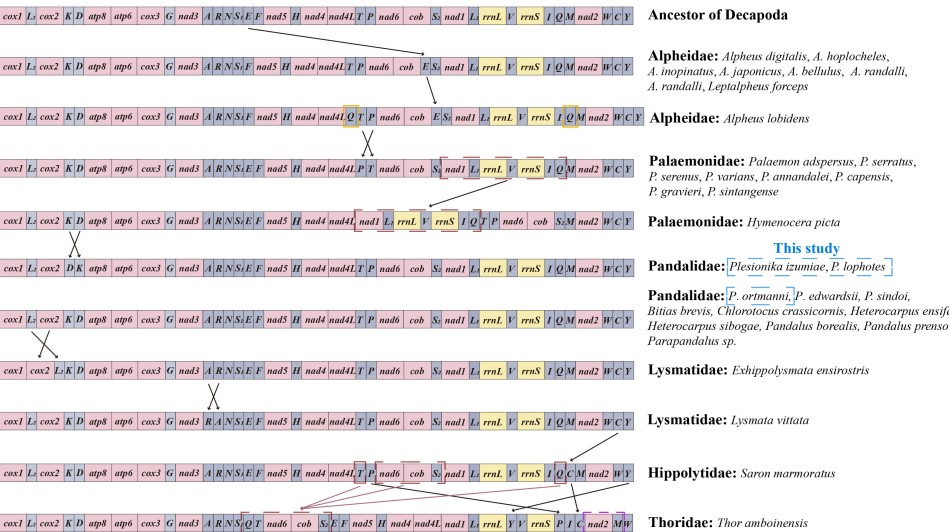

**Figure 6** Linear representation of the mitochondrial gene arrangement of the ancestral mitogenome of pancrustaceans and Caridea species. In this study, the three newly sequenced species are marked with blue box.

(Weber, 1795) species were found to have the translocation of two tRNA genes, with *trnP* or *trnT* involved in the translocation (*Shen et al., 2009*). A novel order was observed in the mitochondrial gene orders (MGOs) of *Hymenocera picta*, with the gene fragment (*ND1-trnL1-16S rRNA-trnV -12S rRNA-trnI-trnQ*) being relocated from downstream of *trnS2* to a position downstream of *ND4L* (*Ye et al., 2021*). In Lysmatidae (Dana, 1852), both *Exhippolysmata ensirostris* and *Lysmata vittata* experienced gene translocation. Specifically, the *trnL2* of *E. ensirostris* was translocated and inversed with *COII* (*Ye et al., 2021*), and the *trnA* of *L. vittata* translocates and inverses with *trnR*. The *trnC* gene in *Saron marmoratus* of the Hippolytidae family was relocated from downstream of *trnW* to a position downstream of *trnQ*. Moreover, the MGOs of *Thor amboinensis* in Thoridae (Kingsley, 1878) have undergone significant changes. Specifically, the *trnQ*, *trnT*, and a gene fragment comprising (*ND6-Cytb-trnS2*) were rearranged from the downstream of *trnI*, *ND4* and *trnP*, respectively, to form a new gene fragment (*trnQ-trnT-ND6-Cytb-trnS2*) downstream of *trnS1*. Additionally, the *trnP* was moved from downstream of *trnT* to the position downstream of *12S rRNA*, the *trnC* was relocated from the downstream of *trnW* to the position downstream of *trnI*, the *trnM* translocated and inverted with *ND2*, and the *trnY* was relocated from the downstream of *trnC* to the position downstream of *16S rRNA*.

In this study, we report the first instances of gene rearrangement in Pandalidae, as observed in the mitogenomes of two newly sequenced *Plesionika* species (*P. izumiae* and *P. lophotes*). This discovery revealed two distinct gene arrangement patterns in the genus *Plesionika*, thereby highlighting the non-conservative nature of gene arrangement in the Pandalidae family. as mitochondrial genomic data for the Pandalidae continue to increase, gene arrangement is expected to yield important evolutionary information that may assist in

inferring the phylogenetic relationships within the Pandalidae and between the Pandalidae and other Caridea groups.

## Phylogenetic relationships

In this study, we constructed a phylogenetic tree of Caridea using the sequences of 13 PCGs in mitogenomes. The analysis included 82 caridean species, focusing on *P. ortmanni*, *P. izumiae*, and *P. lophotes*, with *H. sinensi* and *H. japonica* serving as outgroups. The topological structures of the phylogenetic tree reconstructed using two methods are identical (Fig. 7), but there were slight differences in the support values of some of the clade branches. The support values of BI were generally higher than ML, with the majority of the nodes having a support value of 1. On the other hand, the support values of ML, except for one node with a support value of 53 (the node between *Alpheus japonicus* and *Alpheus randalli*), were between 78 to 100 for all other nodes. The phylogenetic tree analysis demonstrated that all families within Caridea exhibit monophyly. Strong monophyly was exhibited by the families Pandalidae, Thoridae, Lysmatidae, and Hippolytidae, which clustered together forming a single clade. Acanthephyridae and Oplophoridae were closely related in phylogenetic relationship, forming a sister group, and subsequently clustered with Alvinocarididae. A large clade was then formed by these three families with Nematocarcinidae. Additionally, the families of Alpheidae and Palaemonidae were also found to be closely related, forming a sister group relationship.

The phylogenetic tree revealed that these three newly sequenced species within the family Pandalidae did not cluster together. Phylogenetic analysis showed that *P. izumiae* and *P. lophotes* were closely related and formed a distinct branch, while *P. ortmanni*, *P. sindoi* and *P. edwardsii* clustered together on another separate branch. At the genus level, we observed that the genera *Plesionika* and *Heterocarpus* within the family Pandalidae were not monophyletic. Two distinct clades were formed by the five species of the genus *Plesionika*, while *Parapandalus sp.* was grouped with the genus *Heterocarpus*.

## DISCUSSION

It was revealed by a comparison of the base content of the mitochondrial genomes of three *Plesionika* species that a higher AT base content than CG was exhibited by all three genomes, a phenomenon commonly observed in the mitochondrial genome sequences of decapods (*Sun et al., 2020*; *Zhu et al., 2021*; *Wang et al., 2019*). The *trnS1* gene of the *P. ortmanni* lacked the DHU arm, preventing it from forming a typical cloverleaf structure. The loss of the DHU arm in *trnS* gene has been reported in most mitochondrial genome studies of Caridean shrimp (*Ye et al., 2021*; *Shen et al., 2012*). Selective pressure analysis was conducted separately on the PCGs of nine species in the Pandalidae family, and the results showed that the Ka/Ks values of all 13 PCGs were less than 1. *Zhu et al. (2021)* previously performed a selective pressure analysis on species from Palaemonidae family within Caridea and also concluded that the Ka/Ks values of all 13 PCGs were less than 1.

The phylogenetic tree results indicated that the genera *Plesionika* and *Heterocarpus* within the family Pandalidae were not monophyletic. This finding is consistent with a previous study by *Wang et al. (2021)* and *Liao et al. (2019)*, who used mitogenome genes or

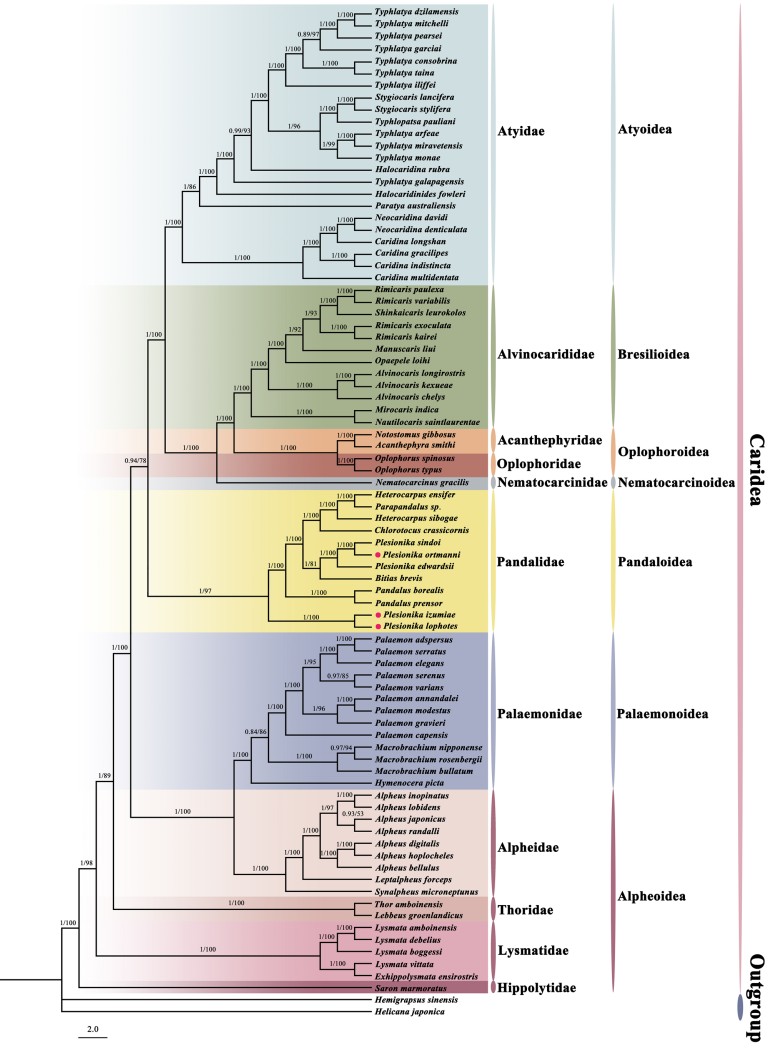

**Figure 7** **The phylogenetic tree based on 13 PCGs was inferred using Bayesian inference (BI) and maximum likelihood (ML) methods.** The number at each clade is the bootstrap probability and the three newly sequenced species are marked with red dots.

nuclear genes to construct phylogenetic trees and examine the phylogenetic relationships within the family Pandalidae. Support for the genera *Plesionika* and *Heterocarpus* being polyphyletic was provided by their results. The five *Plesionika* species included in our study were also assigned to two different clades in their results. Additionally, several studies based on partial mitochondrial sequences (*COI* and *16S rRNA*) have supported the non-monophyly of *Plesionika* (*Silva et al., 2013*; *Chakraborty et al., 2015*; *Chakraborty & Kuberan, 2021*). This result may be attributed to morphological differences among species within the genus *Plesionika*, especially the distinct asymmetry of the second pereiopod in *P. izumiae* and *P. lophotes* compared to other *Plesionika* species. At the family level, Pandalidae exhibits strong monophyly, which is consistent with previous research findings (*Sun et al., 2020*; *Ye et al., 2021*; *Wang et al., 2021*; *Chak, Barden & Baeza, 2020*; *Sun, Sha &*

*Wang, 2021*; *Cronin, Jones & Antonio, 2022*). Regarding phylogenetic relationships among families, the families Acanthephyridae, Oplophoridae, and Alvinocarididae are closely related and form sister groups; similarly, the families Alpheidae and Palaemonidae are also closely related and form sister groups. These results are consistent with previous phylogenetic studies (*Chak, Barden & Baeza, 2020*; *Sun, Sha & Wang, 2021*; *Cronin, Jones & Antonio, 2022*). While our phylogenetic tree topology is consistent with previous research, there are some differences. Specifically, a conflict is noted between our results and those suggested by *Li et al. (2011)*, who proposed that Atyidae represent basal lineages within Caridea, a conclusion based on the analysis of five nuclear genes. Similarly, an inference was made by *Bracken, De Grave & Felder (2009)* that Atyidae constitute basal lineages within Caridea, an assertion based on the analysis of both mitochondrial and nuclear genes. Furthermore, it was found by *Li et al. (2011)* that members of the families Palaemonidae and Hippolytidae did not constitute monophyletic groups. The study suggested that members of Hymenoceridae and Gnathophyllidae were clustered within the Palaemonidae clade and that a close relationship was shown between *Lysmata amboinensis* of Hippolytidae and *Janicea antiguensis* of Barbouriidae. However, according to the latest records from WoRMS, both Hymenoceridae and Gnathophyllidae have been updated to Palaemonidae (*De Grave, Fransen & Page, 2015*), and *L. amboinensis* has also been corrected from Hippolytidae to Lysmatidae (*De Grave & Fransen, 2011*). Therefore, our study supports that Palaemonidae and Hippolytidae are monophyletic groups. This highlights the importance of incorporating molecular techniques into species identification and classification, as they demonstrate the limitations of past morphology-based species taxonomy.

It has been suggested by some scholars in previous studies that mitochondrial gene arrangement could serve as a new molecular marker to assist in phylogenetic analysis (*Zhang et al., 2019*; *Tan et al., 2018*; *Wang, Huang & Zou, 2019*). Some scholars have also recognized the potential of mitochondrial rearrangement as a "super" feature for estimating arthropod phylogenetic (*Boore, Lavrov & Brown, 1998*; *Dowton & Austin, 1999*; *Dowton, Castro & Austin, 2002*; *Tan et al., 2017*). The relationship between mitochondrial gene arrangement and phylogenetics in Caridea was further analyzed in our study. The phylogenetic tree showed a very clear correlation between both gene arrangement and phylogenetic, and the species having gene rearrangement within each family were clustered together. In the Pandalidae family, the newly sequenced species *P. izumiae* and *P. lophotes* exhibited the same gene rearrangement and were closely related in the tree, while three *Plesionika* species (*P. sindoi*, *P. edwardsii*, and *P. ortmanni*) with the ancestral gene order formed a separate branch. This suggests that the polyphyly phenomenon of *Plesionika* may be associated with differences in gene order. In Alpheidae, both the *Alpheus* and *Leptalpheus* (Williams, 1965) genus underwent gene rearrangement and were clustered together with high support. In the Palaemonidae family, species of the *Palaemon* genus were clustered together. Except for *Palaemon modestus*, the same gene rearrangement was undergone by all other species within Palaemon. Conversely, *H. picta*, experiencing a different gene rearrangement, was positioned alone in a separate cluster. In Lysmatidae, the species with gene rearrangement, *L. vittata* and *E. ensirostris*, were also clustered together. Overall, the six families (Pandalidae, Palaemonidae, Alpheidae, Thoridae, Lysmatidae, Hippolytidae)

with gene rearrangements were clustered at the base of the Caridea phylogenetic tree, while the five families sharing the same gene order pattern formed a terminal clade. The results of this study show that there is a certain correlation between the phylogenetics of Caridea and the sorting of mitochondrial genes, but additional mitogenomes data are needed to support this result and further investigate their relationship.

## CONCLUSIONS

We sequenced the complete mitogenomes of three *Plesionika* species and analyzed the basic characteristics of these mitogenomes. It was found that these genomes were relatively similar in terms of size, nucleotide composition, and codon usage preference, but exhibited slight structural differences. Additionally, all 13 PCGs in the 12 species of the family Pandalidae underwent purifying selection, with the *COI* gene experiencing the highest selection pressure, indicating its suitability as an optimal molecular marker for species identification and phylogenetic studies within the Pandalidae. Furthermore, gene rearrangements in the Pandalidae were observed for the first time, translocation of two tRNA genes, *i.e.,* *trnP* or *trnT*, were found in the two newly sequenced *Plesionika* species –*P. izumiae* and *P. lophotes*. Phylogenetic analysis revealed a high level of monophyly within the family Pandalidae, but the genus *Plesionika* appeared to be polyphyletic. By combining the results of gene rearrangements and phylogenetic analysis, a correlation was discovered between the phylogenetics of Caridea and the arrangement of mitochondrial genes. Families that underwent gene rearrangements were located at the base of the Caridea phylogenetic tree, while families without gene rearrangements clustered together at the terminal branch of the phylogenetic tree. This study provides extensive information regarding the mitogenomes of *Plesionika*, laying a solid foundation for future research on genetic variation, systematic evolution, and breeding of *Plesionika* using mitogenomes.

### Funding

This article was financially supported by the Project of Bureau of Science and Technology of Zhoushan (No. 2021C21017) and the National Key R&D Program of China (2019YFD0901204). The funders had no role in study design, data collection and analysis, decision to publish, or preparation of the manuscript.

### Grant Disclosures

The following grant information was disclosed by the authors:
Project of Bureau of Science and Technology of Zhoushan:  2021C21017.
National Key R&D Program of China: 2019YFD0901204.

### Competing Interests

The authors declare there are no competing interests.

## Author Contributions

- Yuman Sun conceived and designed the experiments, performed the experiments, analyzed the data, prepared figures and/or tables, and approved the final draft.
- Jian Chen conceived and designed the experiments, performed the experiments, prepared figures and/or tables, and approved the final draft.
- Xinjie Liang performed the experiments, analyzed the data, prepared figures and/or tables, and approved the final draft.
- Jiji Li analyzed the data, prepared figures and/or tables, and approved the final draft.
- Yingying Ye conceived and designed the experiments, authored or reviewed drafts of the article, and approved the final draft.
- Kaida Xu analyzed the data, authored or reviewed drafts of the article, and approved the final draft.

## Data Availability

All mitogenome sequences data were deposited in GenBank with accession number OP650932 (*Plesionika ortmanni*) (https://www.ncbi.nlm.nih.gov/nuccore/OP650932), OP650933 (*Plesionika izumiae*) (https://www.ncbi.nlm.nih.gov/nuccore/OP650933) and OP650934 (*Plesionika lophotes*) (https://www.ncbi.nlm.nih.gov/nuccore/OP650934).

## Supplemental Information

Supplemental information for this article can be found online at http://dx.doi.org/10.7717/peerj.17314#supplemental-information.

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
