# Peer review of "Sequence comparison of the mitochondrial genomes of Plesionika species (Caridea: Pandalidae), gene rearrangement and phylogenetic relationships of Caridea"

_PeerJ, doi:10.7717/peerj.17314_

## Round 0.1 · original submission · Minor Revisions

I agree with the comments and suggestions of the two reviewers, that this manuscript warrants only minor revision.

**Language Note:** The review process has identified that the English language must be improved. PeerJ can provide language editing services - please contact us at copyediting@peerj.com for pricing (be sure to provide your manuscript number and title). Alternatively, you should make your own arrangements to improve the language quality and provide details in your response letter. – PeerJ Staff

Reviewer 1 ·

Basic reporting

The overall language of the manuscript is understandable. However, professional proofreading service might be necessary as there are still numerous instances where grammar and tenses errors can be found. For example, line 49, mitogenomes is misspelled. Line 46-48, it should be “Compared with the ancestral Caridea, translocation of two tRNA genes, i.e., trnP or trnT, were found in the two newly sequenced Plesionika species – P. izumiae and P. lophotes.”
The writing style of line 82-86, i.e., Bracken et al. ….. and then ends with (Bracken et al., 2009) is incorrect. It should be “Bracken et al. (2009) support the ….”. Another example is at line 109, “Liao et al. (Liao et al., 2019)”, it should be “Liao et al. (2019)”. Please revise throughout the manuscript.
Why is authority of the family ‘Pandalidae’ (line 91) mentioned, but not for other families or taxa? If the authors decide to include authority for taxon, then all taxa, when first mentioned, should have authorities as well.
Line 91-96: are all these biology and lifestyles mentioned exclusively found in Pandalidae?
Line 112-113: Which are deep-water species which are shallow-water species?


Specific comments are found below (not inclusive of all, thus proofreading is highly recommended):
Line 62-64: This sentence is grammatically incorrect.
Line 74: Why mitochondrial genomes is not abbreviated to ‘mitogenomes’, but ‘mitogenome’?
Line 107: This sentence is grammatically incorrect.

Experimental design

The methodology is not clearly described. Methodology should be as clear as possible to ensure reproducibility. Some examples are mentioned below:
Line 127-128: It will be beneficial if the authors describe what are the morphological differences between the three species, although they were identified by experts.
Line 129-130: please describe in detail the salt extraction procedure.
Line 135: Should this be ‘mitochondrial genomic DNA’ or ‘total genomic DNA’?
Line 174-178: This sentence should be separated into two.
Line 176-177: Should be ‘…from Brachyura served as outgroups,…’
Line 174: why is “Sequences” in capital letter?

Table 1. What do you mean by ‘species date’?

Validity of the findings

The findings are justifiable and valid, with sufficient depth. Conclusions are well supported by the findings provided. The slight structural differences in the three mitogenomes of Plesionika were nicely discusssed.

Reviewer 2 ·

Basic reporting

no comment

Experimental design

The authors of this manuscript have done intensively in all standard protocols of mito-genome analysis. The three representative taxa were selected together with several previous deposited mito-genome sequences for analysis. All results are consisted with the protocol usage.

Validity of the findings

no comment

Additional comments

The manuscript is well prepared and the main idea very clear and straightforward for mitogenome descriptive analysis. From my point, this manuscript is deserve to be publish in PeerJ after minor corrections. The short story of phylogenetic relationship of selected taxa/some taxonomic problems found within family Pandalidae might be more mentioned in introduction and discussion to state the missing gap and application of this study result for further work.
All my comments and corrections are directly incorporated into the manuscript to help the authors and improve the quality of the research. The format standardization, and other minor points still deserves a minor revision. The adjustment of some figures and table are required to improve the clarity. Because I'm not native English user so my comments are not intense in grammatical error however, I do recommend author to refine their idea and clarity of this manuscript again.

Annotated reviews are not available for download in order to protect the identity of reviewers who chose to remain anonymous.

---

## Round 0.2 · Minor Revisions

The information, methods and results presented in the study are solid. However, I have to remind the authors that there are still a lot of grammatical errors found throughout the manuscript, below are just some of the examples:

1. "In Palaemonidae, 9 Palaemon (Weber, 1795) species were found the translocation of two tRNA genes, wherein trnP or trnT occurred translocation." This sentence is grammatically incorrect and this will confuse the readers.

2. "Compared with the ancestral Caridea, the MGOs of the newly sequenced P. ortmanni and the other 9 Pandalidae species from GenBank (i.e. Bitias brevis, Chlorotocus crassicornis, Heterocarpus ensifer, Heterocarpus sibogae, Pandalus borealis, Pandalus prensor, Parapandalus sp., P. edwardsii, P. sindoi) remained consistent with the ancestral gene order." When you start a sentence with "compared with...", a comparison is generally expected, but this sentence represents similarity instead of comparison...

3. "Conversely, the two newly sequenced Plesionika species (P. izumiae and P. lophotes) have a translocation, for which the gene order is trnK - trnD instead of trnD - trnK.". This sounds like a direct translation from Mandarin. In English, translocation is a noun. This sentence is grammatically incorrect.

The above three are just examples, there are numerous more throughout the manuscript. Hence, I would strongly urge the authors to seek professional proofreading services.

In addition, the citation format of "Silva et al. (2013) (Silva et al., 2013)" is incorrect. 'Silva et al. (2013)' is already sufficient, there is no need for a redundant citation of (Silva et al., 2013). Please check throughout the manuscript for similar mistakes.

**Language Note:** The Academic Editor has identified that the English language must be improved. PeerJ can provide language editing services - please contact us at copyediting@peerj.com for pricing (be sure to provide your manuscript number and title). Alternatively, you should make your own arrangements to improve the language quality and provide details in your response letter. – PeerJ Staff

---

## Round 0.3 · accepted · Accept

I thank the authors for following through the comments and suggestions. The current version of the manuscript is now ready for publication.